# The Composition and Function of Intestinal Microbiota Were Altered in Farmed Bullfrog Tadpoles (*Aquarana catesbeiana*) during Metamorphosis

**DOI:** 10.3390/microorganisms12102020

**Published:** 2024-10-05

**Authors:** Xiaoting Zheng, Qiuyu Chen, Xueying Liang, Vikash Kumar, Alfredo Loor, Hongbiao Dong, Chang Liu, Jinlong Yang, Jiasong Zhang

**Affiliations:** 1Key Laboratory of South China Sea Fishery Resources Exploitation & Utilization, Ministry of Agriculture and Rural Affairs, South China Sea Fisheries Research Institute, Chinese Academy of Fishery Sciences, Guangzhou 510300, China; xtzheng1990@163.com (X.Z.); ciiyii1010@163.com (Q.C.); changxun100@163.com (X.L.); donghongbiao@163.com (H.D.); liuchang@scsfri.ac.cn (C.L.); 2College of Fisheries and Life Science, Shanghai Ocean University, Shanghai 201306, China; jlyang@shou.edu.cn; 3Aquatic Environmental Biotechnology and Nanotechnology (AEBN) Division, ICAR–Central Inland Fisheries Research Institute (CIFRI), Barrackpore 700120, India; kumar.vika.vikash2@gmail.com; 4Faculty of Maritime Engineering and Marine Sciences (FIMCM), Escuela Superior Politecnica del Litoral (ESPOL), Guayaquil 09015863, Ecuador; alfgloor@espol.edu.ec; 5Sanya Tropical Fisheries Research Institute, Sanya 572018, China

**Keywords:** *Aquarana catesbeiana*, tadpole, intestinal microbiota, metamorphosis

## Abstract

The bullfrog *Aquarana catesbeiana* is one of the main farmed frog species in China, with a low overall survival of farmed bullfrogs from hatching to harvest since bullfrog tadpoles are fragile during the metamorphosis period. The intestinal bacterial community can play crucial roles in animal development; however, little is known about the alteration of the gut microbial community of *A. catesbeiana* during metamorphosis. The present study used 16S rRNA amplicon sequencing to investigate the intestinal bacterial community in *A. catesbeiana* at four distinct developmental stages. Moreover, we determined the bullfrog’s body morphological parameters and the intestine histology at different developmental stages. The results showed a reduction in the total length and snout–vent length of *A. catesbeiana* during metamorphosis. The intestinal microbial composition of *A. catesbeiana* exhibited variation throughout the process of metamorphosis. The terrestrial stage showed shifts in the bacterial composition compared to the aquatic stages, including a reduction in *Bacteroidetes* and an increase in *Firmicutes*. Furthermore, the presence of *Prevotella*, *Bifidobacterium*, *Leucobacter*, *Corynebacterium*, *Bulleidia*, *Dorea*, *Robinsoniella*, and *Clostridium* in *A. catesbeiana* metamorphosis appears to be mainly related to the host’s epithelial cells’ height and total body mass. The results indicated that the intestinal microbial composition changed with the bullfrog–tadpole metamorphosis. The genera of *Prevotella*, *Bifidobacterium*, *Leucobacter*, *Corynebacterium*, *Bulleidia*, *Dorea*, *Robinsoniella*, and *Clostridium* might be potential probiotics.

## 1. Introduction

The bullfrog *Aquarana catesbeiana* was first introduced into mainland China from Cuba in 1962 as a food source due to its high protein provision, soft texture, ease of farming, and high economic value. Since 2000, the bullfrog farming industry in China has experienced rapid growth [1]. In 2021, the production of bullfrogs had reached 600,000 tons, with an economic value of USD 11.2 billion [2]. However, the overall survival rate of farmed bullfrogs from their hatching to harvest is approximately 10%, as bullfrog tadpoles are fragile during the metamorphosis period.

During the metamorphosis stage, the bullfrog tadpoles undergo significant changes in their internal organs, drastically transforming from aquatic larvae to terrestrial adults. According to Gosner’s research [3], the hind limb bud is absent in the larvae at the G25 stage. The metatarsal tubercle appears in the base of the first toe at the G38 stage. Forelimbs emerge at the G42 stage, which is the metamorphic climax. Complete metamorphosis occurs at the G46 stage, when the animal fully develops its limbs and resorbs its tail. During the bullfrog–tadpole metamorphosis, the tadpole first resorbs its tail and gills, which are then replaced by fully developed external appendages and lungs to adapt to a terrestrial lifestyle. As a result, their diets change significantly, with the larvae preferring plant material while the adults are mainly carnivorous. Due to their omnivorous preference, the larvae have evolved a diverse intestinal morphology, ranging from a long, double–spiral intestine to a short intestine in adults [4].

The intestinal microbiota of bullfrogs plays a crucial role in maintaining bullfrogs’ health and undergoes a complex interplay during their metamorphosis [5,6]. From birth, the microbiota co–develops with the host and aids in enhancing the epithelium, regulating immune responses, modulating the energy balance, and influencing the development and behavior of the host [7]. However, the diversity and composition of the microbial community are affected by the host’s genome, nutrition, and lifestyle. Despite this, there are limited studies on bullfrog tadpoles during metamorphosis, which makes it crucial to understand the shifts in the intestinal bacterial community during this process.

The main objective of this study was to determine the diversity and composition of intestinal bacteria in *A. catesbeiana* during bullfrog–tadpole metamorphosis by sequencing the 16S rRNA gene. This study also aimed to investigate the changes in morphological traits and intestinal histology during this process. The results of this study will be crucial to understanding the transition of gut microbial communities in the bullfrog’s life cycle. Additionally, the findings will aid in selecting potential probiotics to enhance the metamorphosis rate.

## 2. Material and Methods

### 2.1. Animals

The bullfrog *A. catesbeiana* specimens were raised at the Guangzhou Shengshi Tangfeng Fishery Co., Ltd. (Guangzhou, China). The embryos were obtained from sexually mature frogs by natural oviposition in March and allowed to hatch in nylon mesh tanks (80 cm × 80 cm × 70 cm, 80 μm) with water at a 30 cm depth without any manipulation. The tadpoles were reared from the embryos and they underwent metamorphic development in plastic tanks (3 m × 5 m × 0.6 m, water depth 0.3 m) using a recirculating aquaculture system. All the larvae were fed identically based on their developmental stages. The determination of the developmental stages of the larvae of *A. catesbeiana* was conducted based on Gosner’s studies [3]. Larvae at the G25 (the hind limb bud is not present yet), G38 (the metatarsal tubercle emerges at the base of the first toe), G42 (forelimbs appear), and G46 (the tail is completely resorbed and the limbs are fully developed) stages were collected. All the animals were starved for one day before measuring morphology and collecting gut samples.

The procedures for collecting and handling the animals were strictly followed, and were provided by the Institution Animal Care and Committee on Laboratory Animal Welfare and Ethics of South China Sea Fisheries Research Institute, Chinese Academy of Fishery Sciences (SCSFRI–CAFS, No. nhdf2024–12) and were consistent with China’s Animal Welfare Legislation guidelines.

### 2.2. Morphology Measurements

Ten larvae were collected randomly from four developmental groups (G25, G38, G42, and G46) to study the morphological changes over the metamorphic stages after being starved for one day. The total body mass (TBM) was measured using an electronic analytical balance, while the total length (TOL), the snout–vent length (SVL), and the body width (BW) were measured as described by Fei et al. [8].

### 2.3. Intestinal Histological Processing

The jejunum samples from 12 tadpoles (G25, G38, G42, and G46, *n* = 3 per stage) were collected to analyze the intestine’s morphological and histological changes during the metamorphosis developmental process. The samples were preserved in 4% formaldehyde and processed for hematoxylin and eosin (H&E) staining. The H&E section and light microscopy images were prepared by Wuhan Sevicebio Technology Co., Ltd. (Wuhan, China).

### 2.4. 16S rRNA Gene Sequencing

Six biological replicates were conducted in each critical developmental group (G25, G38, G42, and G46). In each replicate, three intestines were pooled. The intestine was excised intact from the posterior of the esophagus to the vent, rinsed three times with sterile water, and stored in a 2 mL sterile centrifuge tube. It was immediately frozen in liquid nitrogen and kept at −80 °C for DNA extraction and bacterial community analysis.

The bacterial DNA extraction, PCR amplification, and Illumina NovaSeq sequencing were carried out by Microeco Technology Co., Ltd. in Shenzhen, China. The following steps were followed: First, the total genomic DNA was extracted from the intestinal content samples using the CTAB method. Next, the V3–V4 variable region was PCR amplified with a barcoded fusion primer pair, 341F (5′–CCTAYGGGRBGCAS–CAG–3′) and 806R (5′–GGACTACNNGGGTATCTAAT–3′). Then, the original sequence of all the samples was quality controlled, denoised, spliced, and decimalized using the DADA2 plugin in Qiime2 software to form Operational Taxonomic Units (OTUs). Finally, the OTUs were compared with the Greengenes Database (version 13.8) to obtain species annotation information.

Various methods, including ANCOM, ANOVA, Kruskal–Wallis, LEfSe, and DEseq2, were employed to identify the bacteria, with varying abundances among the samples and groups [9]. The α– and β diversity indices were calculated using the core–diversity plugin within QIIME2 [10]. PLS–DA (partial least squares discriminant analysis) was also used as a supervised model to reveal the microbiota variation among groups, using the “plsda” function in the R package “mixOmics” [11]. Redundancy analysis (RDA) was performed to reveal the association of microbial communities with environmental factors based on the relative abundances of microbial species at different taxa levels using the R package “vegan” [12]. Co–occurrence analysis was performed by calculating Spearman’s rank correlations between the predominant taxa, and the network plot was used to display the associations among taxa. In addition, the potential KEGG Ortholog (KO) functional profiles of microbial communities were predicted with PICRUSt [13]. Unless specified above, the parameters used in the analysis were set at default settings. All data analysis was processed using the online platform Wekemo Bioincloud (https://www.bioincloud.tech) (accessed on 1 September 2024).

### 2.5. Statistical Analysis

The data are presented as the mean ± standard deviation, and statistical analysis was performed using one–way ANOVA (SPSS for Windows, Version 22.0) followed by post hoc Duncan multiple range tests to determine the significant differences between four different, developmental stage groups. A *p*–value of <0.05 was considered statistically significant. The results were presented using GraphPad Prism software (version 7, GraphPad Software, Inc., San Diego, CA, USA) and included all biological repeats and the level of significant difference.

## 3. Results

### 3.1. Morphological Parameters

Based on Figure 1, the total body mass and body width of the bullfrogs during the larval development stages followed a fluctuant model. The total body mass increased to reach a peak in stage G38 (11.6 ± 1.5 g), then decreased in stage G42 (6.2 ± 0.8 g), but increased again in stage G46 (12.2 ± 1.3 g). Similarly, the body width peaked at 1.9 ± 0.1 in stage G38, then steeply decreased in stage G42 (1.3 ± 0.1 cm), and climbed to 1.6 ± 0.1 cm in stage G46. On the other hand, the total length and snout–vent length of bullfrogs during developmental stages showed an approximately parabolic model. The total length and snout–vent length gradually increased, with a maximum length of 10.8 ± 0.4 cm and 3.9 ± 0.2 cm, respectively, in stage G38. However, they subsequently decreased during stages G42–46.

### 3.2. Intestine Histology

The mucosal layer of the jejunum comprised a simple columnar epithelium and the lamina propria was made up of dense connective tissue containing goblet cells (Figure 2A–D). During stages 42–46, an increased number of goblet cells was detected. In comparison to other stages, a higher number of longer, thicker, and curly folds with more complex branches was found in stage G46 (Figure 2D). Furthermore, the height of the jejunum’s epithelial cells in stage G46 was significantly higher than that in stage G38 (*p* < 0.05) (Figure 2E).

### 3.3. Changes in the Intestinal Microbial Composition at the Phylum Level and the Genus Level

Among the four developmental stages, five phyla were the most commonly found in bullfrogs. The phyla were *Fusobacteria* (5.05–66.69%), *Firmicutes* (4.02–74.78%), *Bacteroidetes* (1.06–28.83%), *Proteobacteria* (2.13–13.30%), and *Actinobacteria* (2.58–11.89%). *Verrucomicrobia* were detected at stages G25 (28.4%), G38 (4.2%), and G42 (19.0%), but not at stage G46 (Figure 3A). At the genus level, the first 21 identified taxa are shown in Figure 3B. In stage G25, the primary genera were *u114* (12.2%), *Akkermansia* (26.7%), and *Cetobacterium* (8.2%). The level of *u114* was substantially higher in the stage G38 group at 60.33%, while *Akkermansia* and *Cetobacterium* were at 4.03% and 6.33%. The genus *u114* (29.33%), *Akkermansia* (17.97%), and *Cetobacterium* (11.60%) were the primary genera in the stage G42 group. The genus composition in stage G46 was entirely different from the other three stage groups, with *Weissella* (28.20%), *Clostridium* (22.65%), *Lactococcus* (9.37%), and *Ralstonia* (8.04%) being the main genera.

An LEfSe analysis and cladogram were utilized to identify the biological indicator species in bullfrogs at different developmental stages, using four as the threshold for an LDA score (Figure 3C,D). The stage G42 group showed the most different taxa, with 24 taxa, among which *Caldilinea* was the most highly discriminating taxon. There were significant differences in the intestinal microbiota of bullfrogs at stage G46 amongst the 11 taxa that belonged to the Firmicutes, Actinobacteria, and Proteobacteria groups, respectively. The stage G25 group had nine different taxa, with *Verrucomicrobia* being the most enriched, while the stage G38 group only had five different taxa, with *u114* being by far the most dominant.

The bullfrog samples from the four developmental stages shared 39 OTUs (Figure 3E), with the stage G42 group having the highest number of unique OTUs (1596).

### 3.4. Diversity Analysis of the Intestinal Microbiota

This study compared alpha diversity indices in bullfrogs at four different stages with the Wilcox test (Figure 4A–D). The results showed that the Chao1 index was significantly higher in the stage G38 and stage G42 groups, whereas no significant difference was found in the other stages groups. The observed feature index showed a similar result as the Chao1 index. However, no significant differences existed between groups in the Shannon and Simpson indices. The stage G46 group significantly differed from the other three groups and showed more distance (Figure 4E,F).

### 3.5. Correlations of the Morphological Changes with the Intestinal Microbiota

According to the RDA ranking map, there was a significant correlation between changes in the host morphological parameters and the intestinal microbiota at the genus level (*p* = 0.002, Figure 5A). The correlation heat map (Figure 5B) indicates that among the top 30 genera, *Actinomyces*, *u114*, *Odoribacter*, *Deinococcus*, *c39*, and *Thiothrix* were positively correlated with the total length and the snout–vent length, whereas the rest of the genera were negatively correlated with these two morphological changes. However, stage G38 was the largest for both of these morphological features and had a dominance of *u114* (Figure 3B), which would contribute to the positive correlation of this genus with these two features. On the other hand, the genera of *Prevotella*, *Bifidobacterium*, *Leucobacter*, *Corynebacterium*, and *Bulleidia* were all positively correlated with total body mass, while *Actinomyces* was negatively correlated with it. Moreover, the genera of *Dorea*, *Robinsoniella*, and *Clostridium* were all positively correlated with the epithelial cell height, whereas *C39* was negatively correlated with it.

## 4. Discussion

Throughout the process of tadpoles’ transformation into young frogs, *A. catesbeiana* tadpoles undergo significant physical and physiological changes, as well as a shift in diet from primarily plant–based to carnivorous. The morphological measurements taken in this study demonstrated a significant reduction in the total length and snout–vent length of *A. catesbeiana* during metamorphosis. Similar results were observed in *Rana chensinensis* tadpoles undergoing metamorphosis [14]. These findings suggest that all the changes observed in anuran amphibians are adaptations necessary for transitioning from their aquatic habitat to a terrestrial one.

Intestinal epithelial cells are a crucial component of the intestinal epithelium, where they perform essential functions such as digesting food, absorbing nutrients, and protecting the host from microbial infections [15]. At stage G46, the height of the jejunum epithelial cells was significantly higher than that at stage G38, and more mucosal folds were present. The mucosal folds at stage G46 were longer, thicker, and more complex, with curly branches, which is consistent with the observation in *R. temporaria* tadpoles [16]. Moreover, the number of goblet cells increased during stages 42–46, indicating that the goblet cells adjusted to the changes in dietary habits as the *A. catesbeiana* tadpoles developed.

Stage G46 was found to be separated from the other three stages in the NMDS space, indicating that the G25, G38, and G42 groups shared similar intestinal microbiota. Our research illustrated that the composition of the intestinal microorganisms differed significantly at different developmental stages throughout the metamorphosis of *A. catesbeiana*, which is consistent with the Chinese brown frog (*R. chensinensis*) [14] and the Northern leopard frog (*Lithobates pipiens*) [17]. The G25, G38, and G42 stages take place during the aquatic stage, while stage G46 occurs during the terrestrial stage, so that the quite different microorganisms in stage G46 correlate well with the lifestyle transformation of becoming land–based. The structural changes in intestinal microorganisms observed in our study are likely due to the biological transformations occurring during metamorphosis and the associated lifestyle shift. As metamorphosis induces significant physiological, anatomical, and ecological changes, these transitions reshape the host’s environment, driving shifts in the gut microbiota. Our findings indicate that post–metamorphosis, the microbiota adapt to reflect the host’s evolving nutritional and environmental demands.

This observation is consistent with previous studies, such as Warne et al. [18], which demonstrated that gut microbiota manipulation during critical developmental windows impacts host physiology and disease susceptibility. The dynamic interaction between microbiota and host development suggests microbial shifts not only respond to environmental changes, but also influence physiological adaptations. These shifts, driven by metamorphosis, may enhance the host’s survival and performance in the post–metamorphic stage.

Thus, our hypothesis aligns with Warne et al. [18], supporting a bidirectional relationship between host development and microbiota. Further studies are needed to determine whether specific microbial changes directly affect growth, immunity, and disease resistance, providing insights into gut health management during critical developmental phases.

The G25, G38, and G42 groups exhibited a community dominated by the phyla *Fusobacteria* and *Bacteroidetes*, while the G46 group maintained a community rich in *Firmicutes*. The presence of *Fusobacteria* was higher in the stage G25, G38, and G42 groups than in the G46 group. This trend was observed in the gut of several fish species, such as *Lepomis macrochirus* (82.60%), *Micropterus salmoides* (90.60%) and *Ictalurus punctatus* (94.90%) [19]. In comparative studies of the intestinal microbiota of different mammals, it was found that marine mammals had a considerably higher abundance of *Fusobacteria* than terrestrial mammals [20]. However, while the abundance of *Fusobacteria* detected was at low levels in the gut microbiota of Northern leopard frog adults (0.32%) [17], it was almost completely absent in tadpoles (<0.01%), suggesting different frog species may have different microbiota Hence, the function of *Fusobacteria* in aquatic and terrestrial animals needs to be explored further.

It was found that the *Bacteroidetes* phylum was more abundant in terrestrial animals (34.00–39.40%) when compared to aquatic fish samples (4.85–19.03%) [21,22,23]. Interestingly, the presence of a *Bacteroidetes*–rich gut community was observed in adult frogs of *L. pipiens* (22.82%) but not in tadpoles (2.43%) [17]. The pre–metamorphosis process (G25, G38 and G42) of *A. catesbeiana* is the aquatic stage, while the metamorphosis process (G46) of *A. catesbeiana* is the terrestrial stage. In the present study, it was observed that the abundance of *Bacteroidetes* was higher in the pre–metamorphosis process (G25, G38, and G42, 7.46–28.83%) compared to the metamorphosis process (G46, 1.06%), which is contrary to the observation made in *L. pipiens*.

During the G46 stage, the number of *Bacteroidetes* decreased in *A. catesbeiana* bullfrogs, which *Firmicutes* replaced. This change resulted in a higher ratio of *Firmicutes*–to–*Bacteroidetes*, which could enhance the efficiency of calorie uptake from the food they consume [24]. The major components of these *Firmicutes* were *Weissella* (28.20%) and *Clostridium* (22.65%), which are believed to support digestion and nutrient acquisition during the G46 metamorphosis stage, though further investigation is required to determine the exact role of *Weissella* and *Clostridium*. Nonetheless, the increased *Firmicutes*–to–*Bacteroidetes* ratio suggests that metamorphosis and lifestyle changes can significantly impact the microbial community in *A. catesbeiana* bullfrogs.

According to the RDA analyses and Spearman correlation, changes in the morphology of the host have distinct effects on the bacterial community in the intestine of *A. catesbeiana*. The bacteria related to *Prevotella*, *Bifidobacterium*, *Leucobacter*, *Corynebacterium*, and *Bulleidia* were all found to be positively correlated with the total body mass. In contrast, the *Dorea*, *Robinsoniella*, and *Clostridium* genera were positively correlated with the epithelial cell height. The measured factors of both the epithelial cell height and the total body mass were found to be clustered together, indicating that the intestinal epithelial cells play a vital role in the digestion of food and absorption of nutrients, thereby contributing to increasing the host’s total body mass [25]. Furthermore, the increase in total body mass was accompanied by a simultaneous increase in the *Prevotella*, *Bifidobacterium*, *Leucobacter*, *Corynebacterium*, and *Bulleidia* genera. On the other hand, an increase in the genera of *Dorea*, *Robinsoniella*, and *Clostridium* was accompanied by an increase in epithelial cell height. These bacterial genera have the potential to be developed as probiotics in the future.

## 5. Conclusions

This present study found that the gut microbiota of the *A. catesbeiana* bullfrog were affected during metamorphosis, which involved changes in the host’s morphological parameters. The gut bacteria found during the bullfrog’s four developmental stages were separated into two groups based on their phylogenetic composition, indicating a solid connection to the host’s living habitat, from aquatic to land. Moreover, the presence of certain bacteria, such as *Prevotella*, *Bifidobacterium*, *Leucobacter*, *Corynebacterium*, *Bulleidia*, *Dorea*, *Robinsoniella*, and *Clostridium*, during the metamorphosis process seems to be closely related to the height of the host’s epithelial cells and the total body mass. Overall, this present study provides the first insight into the remodeling of the intestinal microbiota of the *A. catesbeiana* bullfrog that occurs during metamorphosis.

## Figures and Tables

**Figure 1 microorganisms-12-02020-f001:**
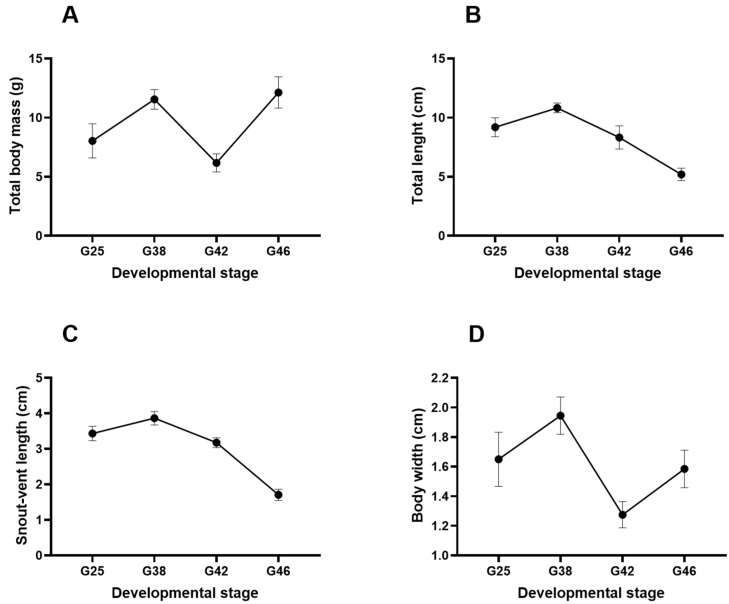
Morphological measurements of *Aquarana catesbeiana* at four different developmental stages (G25, G38, G42, and G46), illustrated as (**A**) total body mass, (**B**) total length, (**C**) snout–vent length, and (**D**) body width. The results are expressed as the mean ± standard deviation, *n* = 10 tadpoles per developmental stage.

**Figure 2 microorganisms-12-02020-f002:**
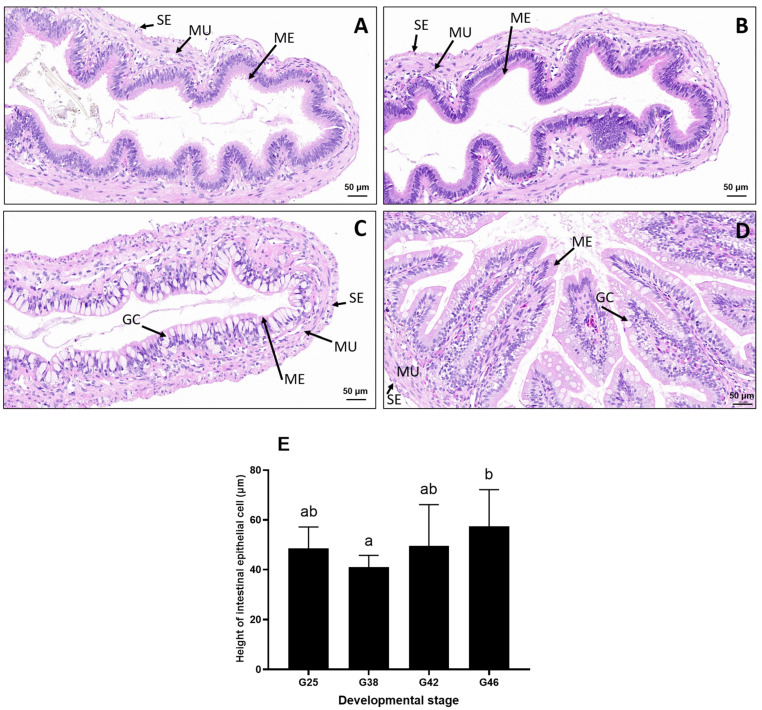
Cross–sections of the jejunum and intestinal morphological measurements of *A. catesbeiana* at four different developmental stages. (**A**) G25 stage. (**B**) G38 stage. (**C**) G42 stage. (**D**) G48 stage. The height of intestinal epithelial cells is also depicted (**E**). The scale for A–D is 50 μm. The bars represent the mean ± standard deviation. The different letters indicate significant differences (*p* < 0.05). GC: goblet cells, ME: mucosal epithelium, MU: muscularis, SE: serosa.

**Figure 3 microorganisms-12-02020-f003:**
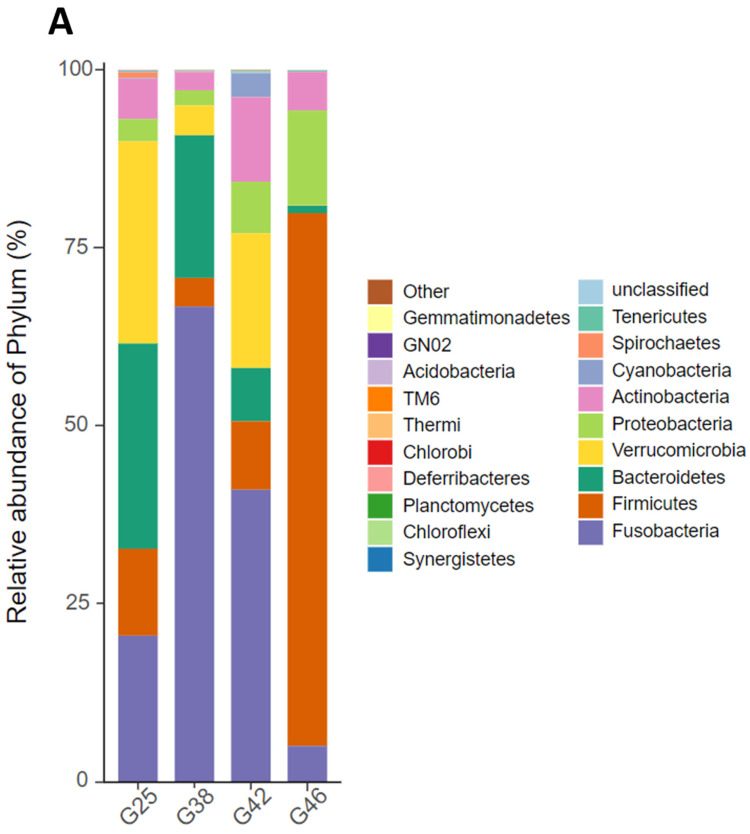
A comparison of the relative abundance of different taxa in four different developmental stages (G25, G38, G42, and G46) of *Aquarana catesbeiana*. (**A**) The relative abundance of taxa at the phylum level. (**B**) The relative abundance of taxa at the genus level. (**C**) Using LEfSe to analyze differences in bacterial taxa at the genus level with an LDA score of >4. (**D**) LEfSe analysis cladogram at the genus level. (**E**) Venn diagram of shared and unique OTUs.

**Figure 4 microorganisms-12-02020-f004:**
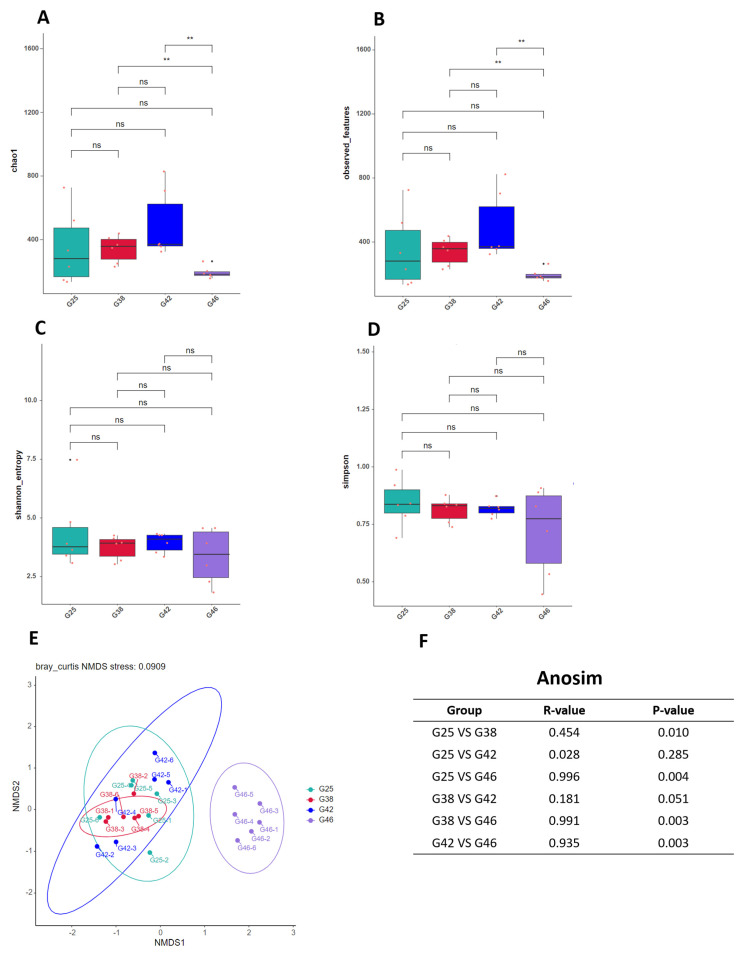
The comparison of alpha diversity and beta diversity among four different developmental stages (G25, G38, G42, and G46) of *Aquarana catesbeiana* based on different indices. (**A**) Chao1. (**B**) Observed OTUs. (**C**) Shannon index. (**D**) Simpson index. ns, no significance, Wilcox test. (**E**) NMDS based on Bray–Curtis distances. (**F**) ANOSIM analysis. Red dots represent six biological replicates. **: *p* < 0.01 (very significant).

**Figure 5 microorganisms-12-02020-f005:**
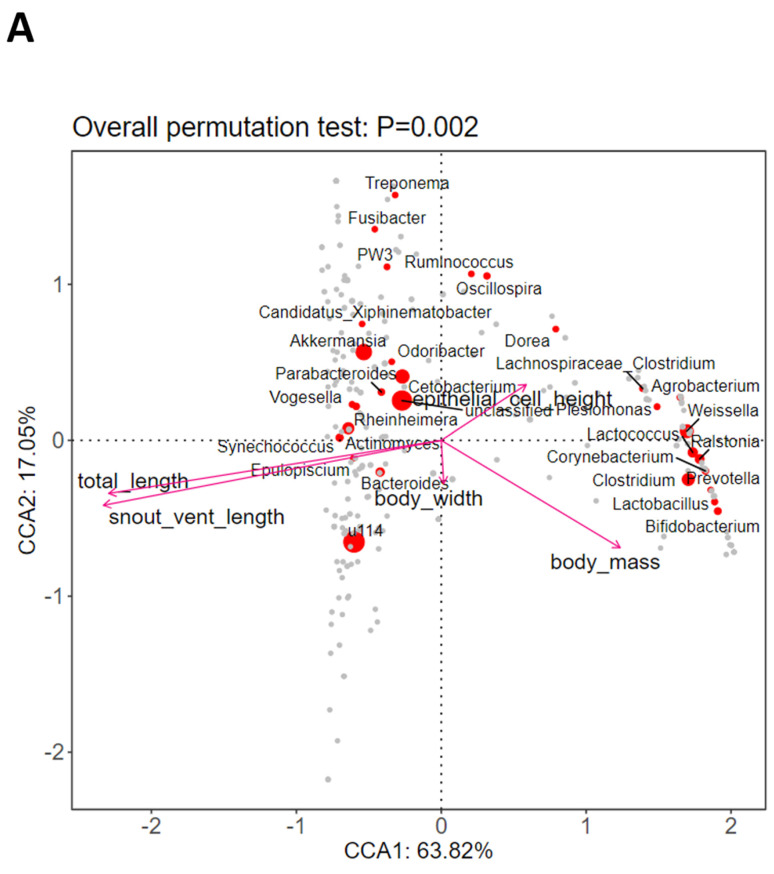
(**A**) RDA correlation tri–plots and (**B**) Spearman correlation between microbiota composition and host morphological parameters (body mass, total length, snout–vent length and body width, and epithelial cell height) at the genus level. The pink arrows indicate variables, and the circles represent the top 30 genera from four developmental stages. The angle between the variables indicates the correlations between microbiota composition and host factors. An acute angle between variables indicates a positive correlation, while a right angle shows no correlation, and an oblique angle represents a negative relationship. The heat map uses different color gradients to indicate the correlation, and the significance is demonstrated as *p* < 0.05 (*), *p* < 0.01 (**, very significant), and *p* < 0.001 (***, extremely significant).

## Data Availability

Dataset available on request from the authors.

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
