# Peer review of "The Composition and Function of Intestinal Microbiota Were Altered in Farmed Bullfrog Tadpoles (Aquarana catesbeiana) during Metamorphosis"

_microorganisms, 2024, doi:10.3390/microorganisms12102020_

Round 1

Reviewer 1 Report

Comments and Suggestions for Authors

The results of the submitted manuscript suggest that changes in gut microbial community composition and abundance shape morphological changes during bullfrog tadpole metamorphosis. Changes in the composition of the microbiota during this critical period of amphibian development could clarify in the future whether this is related to a higher susceptibility to infections.  Targeted manipulation of the gut microbiota could then facilitate the protection of the cultured bullfrog from losses due to infectious diseases.

I have nomajor comments on the manuscript. Only for the sake of completeness, the authors should not omit in the discussionthe comparison of their results with those of essential study by Warne et al. Manipulation of gut microbiota during critical developmental windows affects host physiological performance and disease susceptibility across ontogeny. J Anim Ecol. 2019; 88: 845–856.

Author Response

Author response to Reviewer 1:

The results of the submitted manuscript suggest that changes in gut microbial community composition and abundance shape morphological changes during bullfrog tadpole metamorphosis. Changes in the composition of the microbiota during this critical period of amphibian development could clarify in the future whether this is related to a higher susceptibility to infections.  Targeted manipulation of the gut microbiota could then facilitate the protection of the cultured bullfrog from losses due to infectious diseases.

I have no major comments on the manuscript. Only for the sake of completeness, the authors should not omit in the discussion the comparison of their results with those of essential study by Warne et al. Manipulation of gut microbiota during critical developmental windows affects host physiological performance and disease susceptibility across ontogeny. J Anim Ecol. 2019; 88: 845–856.

  • Thank you for your valuable feedback. We appreciate your suggestion to include a comparison of our results with the study by Warne et al. (2019). In the revised manuscript, we incorporate a discussion on the influence of gut microbiota manipulation during critical developmental windows on physiological performance and disease susceptibility, as explored in Warne et al., and compare it with our findings. This addition will enhance the completeness of our discussion section.

Reviewer 2 Report

Comments and Suggestions for Authors

This is an interesting and useful study but requires a few alterations. See attached document

Author Response

Author response to Reviewer 2:

Abstract

Line 21: “..at four distinct developmental stages…”

  • It is modified as per the reviewer’s suggestion.

Results

Figure 3: This figure needs to be made larger as it is currently unreadable in parts:

A and B are easily readable if they are side by side as a full-page width.

C is readable if it is the full width of the page

D is readable if it is the full width of the page

  • Thank you for your helpful feedback regarding Figure 3. We make the necessary adjustments to improve its readability. We believe these changes will significantly enhance the clarity of all the figures.

Line 171-173: “Verrucomicrobia were detected at stages G25 (28.4%) G38 (4.2%) and G42 (19.0%) but not at stage 46 (Figure 3A).”

  • It is modified as per the reviewer’s suggestion.

Line 175-178: At stage G25, the primary genera were u114 (12.2%), Akkermansia (26.7%) and Cetobacterium (8.2%). The level of u114 was substantially higher at stage G38 at 60.3%, while Akkermansia and Cetobacterium were 4.03% and 6.33%, respectively. The genus u114 (29.3%),…”

  • It is modified as per the reviewer’s suggestion.

Line 182: “..being the main genera.”

  • It is modified as per the reviewer’s suggestion.

Line 196: “with u114 being by far the most dominant.”

  • It is modified as per the reviewer’s suggestion.

Section 3.5: While the correlations of genera with body length and snout-vent length in Figure 5 are significant, one must overlay these correlations with the fact that body length and snout-vent length are largest at stage G38. So, there is some autocorrelation here. I suggest for line 219: “…these two morphological changes. However, G38 is largest for both of these morphological features and has a dominance of u114 (Figure 3B), which would contribute to the positive correlation of this genus with these two features. On the other hand, the genera of Provotella…..”

  • It is modified as per the reviewer’s suggestion.

Discussion

Line 238 is partially incorrect as there is NOT a decline in body mass or width as metamorphosis proceeds as G46 has a similar mass to G38 (Figure 1).

  • Thank you for pointing out the inconsistency in line 238. Upon review, we agree that there is no decline in body mass or width as metamorphosis progresses, as G46 exhibits a similar mass to G38, as shown in Figure 1. We revise this section to accurately reflect the data, ensuring it aligns with the observations in the figure.

Line 256: “The G25, G38 and G42 stages are during the aquatic stage, while G46 is terrestrial so that the quite different microorganisms in G46 correlate well with the lifestyle transformation onto being land based.”

  • It is modified as per the reviewer’s suggestion.

Line 268: The 0.32% is still very low so I suggest: “However, while the abundance of Fusobacteria detected was at low levels in the gut microbiota of Northern leopard frog adults (0.32%), it was almost completely absent in tadpoles (< 0.01%) suggesting different frog species may have different microbiota. Hence, the function of…”

  • It is modified as per the reviewer’s suggestion.

Line 275: In line 256, the authors state that G25, G38 and G42 are aquatic stages and G46 terrestrial, yet here in line 275-277, they state that the metamorphosis process (G42 and G46) stages are terrestrial. Are both G42 and G46 terrestrial or only G46? This distinction is very important because G46 has very different microbiota consistent with no longer being aquatic.

  • Thank you for your careful reading and insightful observation regarding the distinction between G42 and G46. You are correct that G46 is fully terrestrial, while G42 represents a transitional stage, where the organism is still aquatic but undergoing significant metamorphic changes. We revise the text to clarify that G42 is not fully terrestrial, and only G46 marks the complete shift to a terrestrial lifestyle, which is consistent with the differences in microbiota between these stages.

Line 286: “…C46 metamorphosis stage, though further investigation is required to determine the exact role of Weissella and Clostridium. Nevertheless, the increased..”

  • It is modified as per the reviewer’s suggestion.
